# Perceptual-cognitive skills and talent development environments in soccer: A scoping review

Andrew O. Triggs[1,2]*, Joe Causer[2], Allistair P. McRobert[2], Matthew Andrew[3]

1 UCFB Manchester Campus, Manchester, United Kingdom, 2 Research Institute for Sport and Exercise Sciences, Liverpool John Moores University, Liverpool, United Kingdom, 3 Department for Sport and Exercise Science, Manchester Metropolitan University Institute of Sport, Manchester, United Kingdom

* a.o.triggs@2021.ljmu.ac.uk

## Abstract

Perceptual-cognitive skills are a key performance component within professional soccer. Consequently, their role within talent development environments has gained increasing attention. Despite this growing interest, research linking perceptual-cognitive skills to the talent development process remains relatively limited. The present study provided a scoping review examining perceptual-cognitive skills within soccer-specific environments within the last three decades, with a particular focus on outcomes relevant to talent development contexts. Following PRISMA guidelines, specific inclusion and exclusion criteria were set, where 55 studies were included in the final review. Narrative analysis identified key themes in the literature, including visual search behaviours, cognitive capabilities, performance, and methodologies. These themes are discussed with areas for future research identified to support the work of stakeholders in talent development contexts (e.g., coaches, scouts, academy directors), to re-direct future research efforts to further bridge the gap between science and application.

## Introduction

The identification and development of talented youth soccer players who have the potential to progress to professional status is a key objective for soccer academies and national governing bodies [1]. Over recent years, professional clubs have increased investment in their talent identification (TI) and development (TD) programmes, employing multidisciplinary teams including coaches [2], scouts [3], sport scientists [4], performance analysts [5], and more recently, data scientists [6] to enhance their TI and TD processes by creating an environment that educates and supports young players across multiple domains, including technical, tactical, physical, psychological, lifestyle, and well-being aspects of both soccer and personal development [7]. To support these efforts, extensive research has explored key

**Data availability statement:** This is a review paper and so raw data is not present. All relevant data (articles reviewed) are within the manuscript.

**Funding:** The author(s) received no specific funding for this work.

**Competing interests:** The authors have declared that no competing interests exist.

predictors of future expert performance, identifying factors such as physical attributes (e.g., speed; [8,9]), skill (e.g., technical; [10–12]), sociological influences (e.g., developmental activities; [13,14]), psychological characteristics (e.g., motivation; [15]), and chance events (e.g., relative age effect; [16]) (for a review, see [17]). These insights have provided an evidence-based foundation for professional clubs to refine their identification, development, and (de)selection processes, providing opportunities for a more systematic approach within talent pathways [17,18]. However, even with an increase in research which provide opportunities for improved knowledge and the integration of evidence-based practice; principles and frameworks within talent pathways remain largely experience informed [19].

It is well accepted that in combination with high-levels of technical skills (e.g., passing, shooting, dribbling), a future predictor of expert performance in soccer are perceptual-cognitive skills (PCS; [17]). PCS skills are defined as an ability to identify and acquire environmental information that is integrated with existing knowledge, where an appropriate goal-directed response(s) can be selected and executed [20]. PCS are complex but include: *visual search*, which refers to an athletes ability to attend to important/relevant information of the unfolding event(s)/action(s) [21]; *anticipation*, which refers to an athletes ability to predict the event(s)/action(s) that are likely to unfold prior to an event occurring [22]; and *decision-making*, which refers to an athletes ability to use information from the current situation to plan, select and execute an appropriate goal-directed action(s) [23]. Empirical research supports the importance of PCS in distinguishing skilled from less-skilled players. O'Connor et al., [24] demonstrated that elite youth footballers outperform non-selected peers on decision-making tasks, suggesting that PCS are a key differentiating factor in talent programmes. Similarly, Roca et al. [25] found that elite players exhibited superior response accuracy in perceptual-cognitive tasks, reinforcing the notion that cognitive processing underpins soccer expertise. Further evidence from Huijgen et al. [26] highlights that elite youth players outperform sub-elite counterparts in executive functioning tasks. PCS can therefore support soccer players to make faster and more effective decisions [27,28] by identifying and utilising multiple sources of information efficiently in a dynamic game environment [29]. PCS are therefore an important component within talent pathways to support the development of future expert performance. As a result, knowing how to identify, monitor and assess players with high level PCS as well as the understanding of how to develop these skills may seem essential for practitioners aiming to have players equipped for the demands of first team soccer at 18 years of age.

Despite the recognised importance of PCS in soccer performance, research within TD environments have largely overlooked their role in assessing and developing future talent. While links between PCS, soccer performance, and talent progression are evident, there remains a lack of clarity into how these skills are identified, assessed, and developed in TD environments. This gap highlights not only a challenge in translating scientific research into practice but also raises questions about whether current research outputs provide practitioners with actionable guidance. If PCS research is to meaningfully impact TD environments, it must align with the

needs of applied practitioners by offering transferable frameworks, clear assessment methodologies, and practical development strategies. Identifying what is missing in the current body of literature is therefore crucial to bridging the science-to-practice gap.

## Present study

The aim of this review was to examine the current literature on PCS in soccer to identify its role within talent pathways and identifying how it can be used to better support practitioners. Specifically, the review explores how PCS have been investigated, and to what capacity these research findings inform those responsible for TD (e.g., coaches). This review provides an opportunity for researchers to reflect and reframe current research interests surrounding soccer and PCS, ensuring that research outcomes have applicability to TD environments or improve the guidance and frameworks around how practitioners can utilise such evidence informed approaches. If research can be more effectively embedded into TD structures, both player development and club outcomes may benefit, reinforcing the rationale for the growing focus on PCS research.

Given the diverse literature base and methodological approaches used in PCS research in soccer, a scoping review was deemed the most appropriate method to map existing research, identify key themes, and provide recommendations for future PCS research within TD contexts [30–32]. As shown in Fig 1, interest in soccer-specific PCS research has accelerated since the early 2000s (e.g., Williams & Ward, [33]). Accordingly, this review covered research from 2003 to 2024, capturing three decades of evolving inquiry. The principal research question was "what is the extent and nature of research conducted on PCS in soccer that provide applications for TD contexts/practitioners within the last three decades?" To our knowledge, this is the first review to comprehensively examine PCS within soccer and their links with TD.

## Methods

A scoping review was selected over a systematic review as: (1) the research area had not been extensively reviewed; (2) there was a diverse body of literature pertaining to a broad topic; and (3), there were a large range of study designs and methodologies within the literature examining PCS [30–32]. Unlike systematic reviews, which address specific questions to inform targeted interventions or practices, a scoping review provides a comprehensive overview to map key concepts and identify gaps in the literature [34]. This approach allowed for the systematic identification of available research on PCS in soccer and its applicability to TD contexts, thereby guiding future research directions [34]. The methodological

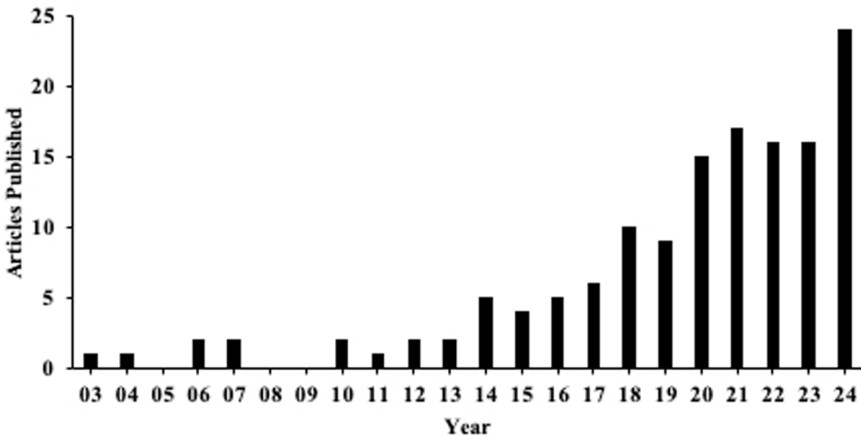

**Fig 1. Number of PCS-soccer specific publications returned from initial inclusion criteria search from 2003 (03) to 2024 (24).**

framework included identifying the research question; identifying relevant studies; selecting relevant studies; charting the data; and summarising, collating the data, and reporting the results [31]. Preferred reporting items for scoping reviews (PRISMA-ScR) checklist items were held (S1 Table) to maintain methodological and reporting quality [35].

## Search strategy

The search strategy was initiated in January 2025 and completed in February 2025. To ensure quality and appropriate coverage, the following electronic databases were used to search for relevant publications; PubMed, PsycInfo, SPORT-Discus, and Web of Science. Additional relevant articles were identified through Google Scholar and hand-picked for inclusion.

The search strategy was designed to capture research on PCS in soccer which included applications relevant for TD contexts/practitioners. Search term one focused on mode of sport and was 'football' OR 'soccer'. Search term two targeted key PCS concepts [36,37], including 'perceptual-cognitive skills', 'game-intelligence', 'scan*', 'anticipation', 'decision-making', and 'visual-search'. To further improve relevance of articles with applications that could be considered for TD practitioners, search term three, 'coach*', was used in addition to the outcomes of search term one and two combined. This term was used to focus on TD, where coaches are one of the main stakeholders involved in such processes [3], and aimed to allow inclusion of articles which included more explicit applications of PCS research outputs. The asterisk used for 'scan*' and 'coach*' acted as a truncation operator and allowed variations of the key word to be found. Search term combinations that returned too many results with lack of specificity to the topic area were removed (Table 1).

## Screening process

The initial search returned 2,702 results across core databases. Articles were title and key word screened for relevance independently by two reviewers, then screened by a third reviewer. Articles meeting the initial inclusion criteria (soccer and PCS specific) were downloaded and indexed using bibliographic manager EndNoteX9 (Clarivate, UK), allowing effective management of the publications. For the publications selected, duplicates were removed ($n = 345$) and abstract

**Table 1. Overview of original search strategy with quantity of results returned.**

| Search Term 1 | Search Term 2 | Search Term 3 | PubMed | SPORTDiscus | PsycInfo | Web of Science | Google Scholar |
|---|---|---|---|---|---|---|---|
| Soccer OR Football | Game Intelligence | | 5 | 20 | 4 | 18 | 1,085 |
| Soccer OR Football | Game Intelligence | Coach* | 1 | 10 | 3 | 9 | 811 |
| Soccer OR Football | Scan* | | Search Discarded | Search Discarded | Search Discarded | Search Discarded | Search Discarded |
| Soccer OR Football | Scan* | Coach* | 19 | 57 | 31 | 61 | 19,900 |
| Soccer OR Football | Anticipation | | 138 | 124 | 92 | 410 | 24.460 |
| Soccer OR Football | Anticipation | Coach* | 10 | 16 | 7 | 23 | 20,430 |
| Soccer OR Football | Decision-making | | Search Discarded | Search Discarded | Search Discarded | Search Discarded | Search Discarded |
| Soccer OR Football | Decision-making | Coach* | 145 | 301 | 104 | 427 | 25,770 |
| Soccer OR Football | Visual Search | | 48 | 55 | 40 | 214 | 10,594 |
| Soccer OR Football | Visual Search | Coach* | 9 | 6 | 4 | 30 | 4,162 |
| Soccer Or Football | Perceptual- Cognitive Skills | | 45 | 65 | 25 | 82 | 8,466 |
| Soccer Or Football | Perceptual-Cognitive Skills | Coach* | 9 | 14 | 2 | 19 | 5,496 |

screening commenced ($n=567$) independently by the same reviewers, where the following inclusion criteria needed to be met: empirical articles; publications from 2003-2024; within soccer; focus on PCS or key terms related to this; contain applications/implications for TD settings or direct applications for applied practitioners (e.g., coaches); participants (male or female, youth or adult) from soccer-teams (any playing level due to broader applicability of research findings) and could include practitioners (e.g., coaches) or players but not include teachers or school students; the article published in English; and had to relate to outfield players/not include analysis of penalty kicks due to its lack of representation of the whole game of soccer (Table 2).

Reviewers utilised a red-amber-green (RAG) rating scale due to its universal simplicity and understanding of the reviewer's status with their decision-making process [38], where any amber rated publications were discussed with additional members of the research team. If a decision could not be made or the decision to include/exclude was not unanimous, the most experienced author made the final decision. Qualitative records from the selected articles were extracted into a single Microsoft Excel spreadsheet for validation and coding. The articles remaining ($n=125$) were full text screened by the same reviewers, utilising the same inclusion/exclusion criteria stated, before agreement checks of included/ excluded articles by two additional reviewers. Amber coded articles followed the same agreement processes as previous. After coding, the selected publications ($n=55$) were appraised for: major foci of the study; participants and contexts; methodology; key findings and applications of findings for TD contexts/practitioners in soccer (see Fig 2 for selection process summary). Despite the application of inclusion/exclusion criteria, the selected studies remained diverse in scope, methodology, and focus, reinforcing the rationale for adopting a scoping review approach [34]. Given the review focus, there were no specific parameters, nor were studies to be analysed regarding the quality of their methodological approach due to the variety and diversity of research encompassing PCS in soccer [31]. Moreover, a formal paper-by-paper critical quality assessment was not undertaken as this is not a key component of scoping reviews [35,39]. Table 3 provides an overview of the reviewed publications.

## Analysis

A narrative analysis was employed to synthesise and interpret the findings from the included studies, identifying patterns, themes, and key messages within the literature on PCS in soccer-specific environments with applications for TD and its practitioners. This approach was particularly appropriate given the diversity of methodologies, terminologies, and frameworks used in PCS research, allowing for a structured yet flexible examination of how these skills have been investigated and applied within the time frame examined. The narrative analysis provided a qualitative framework to compare findings, assess trends, and highlight inconsistencies within the literature. Studies were categorised into key themes, including

**Table 2. Inclusion criteria.**

| Characteristic | Inclusion Criteria |
|---|---|
| Language (Eligibility Criteria 1) | English speaking. |
| Study (Eligibility Criteria 2) | Empirical in nature. |
| Practical Implications (Eligibility Criteria 3) | Must include applications or implications relevant to TD environments in soccer. Studies providing direct recommendations for applied practitioners (e.g., coaches) were also included. |
| Participants (Eligibility Criteria 4) | Participants must be soccer players (male or female, youth or adult) from a team-based environment at any playing level, or practitioners (e.g., coaches, scouts). Studies involving teachers, school/university students or non-soccer athletes were excluded. |
| Area (Eligibility Criteria 5) | The study must investigate perceptual-cognitive skills (PCS) or directly related terms within a soccer-specific context. |
| Time (Eligibility Criteria 6) | Articles must have been published between 2003 and 2024 to capture research developments across three decades. |
| Soccer Specificity (Eligibility Criteria 7) | Use of outfield players and does not utilise the analysis of penalty kicks. |

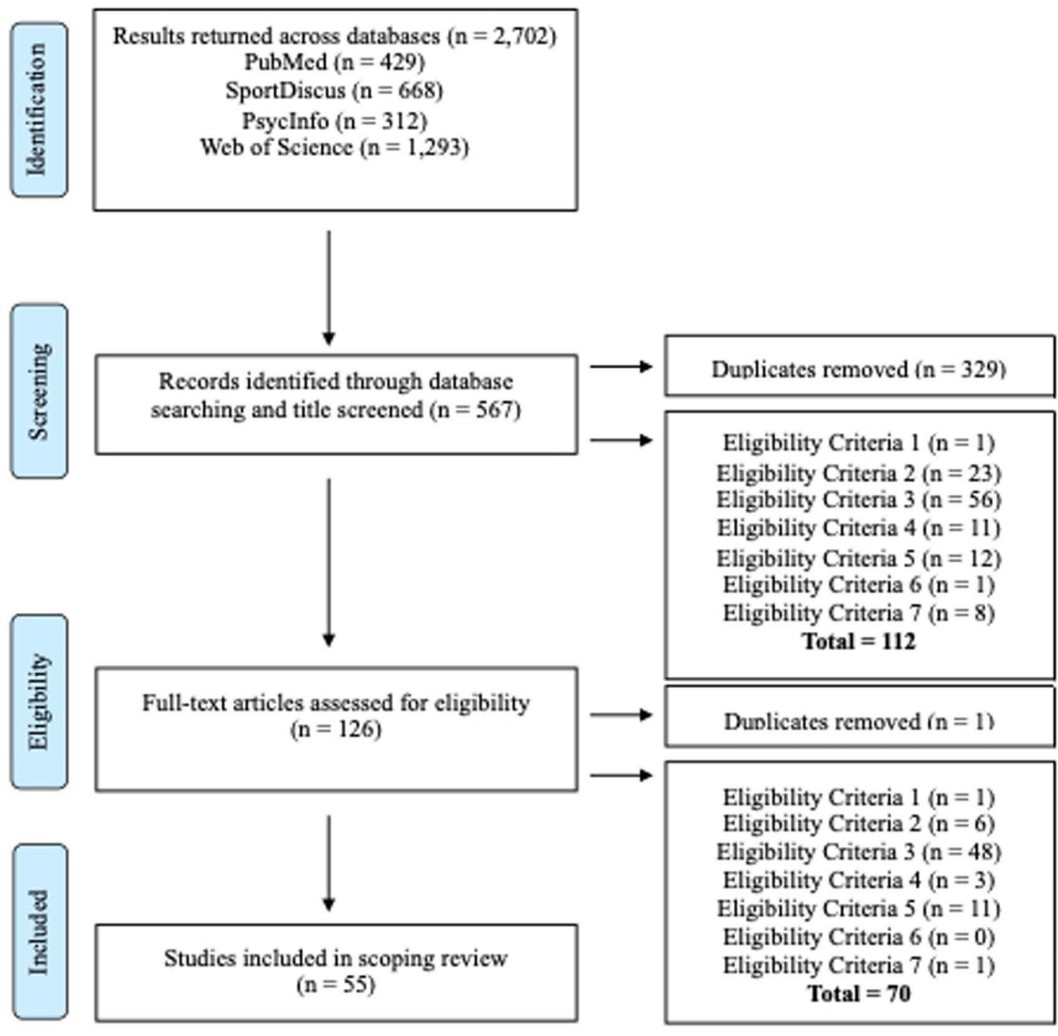

**Fig 2. PRISMA flow chart of process to generate studies for review.**

visual search behaviours, cognitive capabilities, performance, and methodological approaches. This process facilitated a deeper understanding of how PCS research has evolved, to what extent findings have been applied to TD contexts, and where gaps remain between research and practice. Informed recommendations were then developed by the research team with the view to better support integration of research to practice.

## Results and discussion

The aim of the review was to analyse the extent and nature of research conducted on PCS in soccer-specific environments, with applications focussed on TD contexts or applied practitioners since 2003. This section discusses the themes and sub-themes (Table 4) generated from the narrative analysis (visual search behaviours, cognitive capabilities, performance, and methodologies), providing avenues for future research to enhance both research directions and applied practice [1].

**Table 3. Overview of reviewed articles.**

| Authors | Study Aim | Study Sample | Method | Important Results/Findings |
|---|---|---|---|---|
| Aksum et al., [45] | To examine how the scanning behaviour of elite youth soccer players in Europe related to players' performance, according to situational, context-specific, and temporal constraints. | Outfield players (n = 53) from the four teams who reached the finals in the 2018 UEFA European U17 (n = 24) and U19 (n = 29) Championships. | • Data was collected by filming the semi-finals and finals of the 2018 UEFA European U17 and U19 Championships. | • U19 players performed more scans than U17 players.<br>• There is a positive relationship between scan frequency and pass success.<br>• Opponent pressure and pitch position are critical contextual factors that influence scanning behaviour.<br>• Central midfielders and central defenders have higher scan frequencies. |
| Aksum et al., [51] | To explore the scanning behaviour of four elite football midfield players in 11 vs. 11 match play. | Four male central midfield players aged 17–23 who played for two different clubs in the Norwegian Premier League. | • Two 11 vs. 11 matches played: an internal training match and a local third division team. Participants wore eye-tracking devices.<br>• Two of the players were recorded for 20 minutes each, and two players were recorded for 10 minutes each. | • Action undertaken with the ball at the moment of scanning initiation influenced scanning duration.<br>• Playing phase and player-to-ball distance influenced the number of teammates and opponents inside the video frame during scanning.<br>• Very few scans involved fixations.<br>• Scanning duration is not influenced by player-to-ball distance and playing phase. |
| Andrade et al., [82] | To verify the influence of positional role on the decision-making skills of U13 soccer players. | 30, U13 male soccer players, from regional soccer schools (Brazil). | • TacticUP® online platform used to measure decision-making quality and speed. | • No differences in decision-making quality between the different positional roles.<br>• Midfielders made quicker decisions, when compared to defenders and forwards, in situations near and distant from the ball.<br>• Positional role influenced the decision-making time of U13 soccer players. |
| Ballet et al., [66] | To explore the visual behaviour and attentional effort of three football players in specific-role positions in the five seconds before receiving the ball. | A left back, centre midfielder and right winger male football players who play as amateurs in the Portugal National University Championships. | • 11v11 game was used, where 24 game sequences were recorded and analysed with the Tobii Pro eye-movement registration glasses. | • Players acting in different specific positions presented distinctive visual behaviours and attentional effort during a football game. |
| Beavan & Fransen, [91] | To determine if precueing using tactical instructions influences soccer players' decision-making skill in a sport-specific task. | 13 male intermediate soccer players were recruited from several regional (Australia) soccer clubs. | • A decision-making task that was preceded by a video containing either the basic rules of soccer (neutral condition) or specific coaching instructions that were either compatible or incompatible with the correct decision (precue condition). | • Decision-making accuracy was lower and response time was faster in the precue condition.<br>• Fixation location marginally influenced by the compatibility of the precues.<br>• Compatible precues improve sport specific decision-making while incompatible precues hinder performance. |
| Bennett et al., [144] | To investigate the construct and discriminant validity of a video-based decision-making assessment for talent identification in youth soccer. | 328 male youth soccer players from three academy systems: tier one (n = 119) tier two (n = 171) and tier three players (n = 38). An additional 59 youth athletes with no competitive soccer experience in the last five years formed the control group. | • Players completed a video-based decision-making assessment on an iPad, with response accuracy and response time recorded for various attacking situations (2 vs. 1, 3 vs. 1, 3 vs. 2, 4 vs. 3, and 5 vs. 3). | • The video-based decision-making assessment showed construct validity.<br>• Response times were significantly faster in the early and mid-adolescent players.<br>• An overall decline in decision-making performance was observed from the 2 vs. 1 to the 5 vs. 3 situations.<br>• The video-based decision-making assessment lacked discriminant validity as minimal differences between academies were evident for response accuracy and response time. |

*(Continued)*

| Authors | Study Aim | Study Sample | Method | Important Results/Findings |
|---|---|---|---|---|
| Bishop et al., [55] | To identify visual search characteristics that determined superior performance on a soccer-based task and use those characteristics to improve novices' performance of the same task. | Experiment 1–26 male and 14 female participants, with experiences in competitive soccer ranging from complete novice to semi-professional level. Experiment 2 – A total of 46 undergraduate students who were not soccer players. | • Experiment 1–96 still images depicting three oncoming attacking soccer players. Participants task was to predict the direction (left/right) in which they believed the player was about to move by clicking buttons. Eye tracking used to measures saccades.<br>• Experiment 2 – Same method, however, two groups were told to look at the head or ball as quickly as possible before making a decision. | • Sole predictor of decision-making efficiency was the time taken to initiate a saccade to the ball.<br>• There were no between- groups differences for decision-making efficiency with verbal direction. |
| Cardoso et al., [69] | To evaluate how response time in decision making managed by systems 1 (intuitive) and 2 (deliberative) is associated to perceptual-cognitive processes of young soccer players. | 90 male youth soccer players from the youth teams of a Brazilian First Division club. | • Participants wore a mobile-eye tracking system while viewing 11-a-side match play video-based soccer simulations. | • Players with shorter response time in decision making employed more fixations of shorter duration and showed less cognitive effort in the information processing, decision verbalisation and recovery phases.<br>• The faster group also shower a greater number of thought processes associated with planning when compared to players with longer response time in decision making. |
| Casanova et al., [70] | a)To examine the effects induced by physiological workload on gaze behaviour during a 2 vs. 1 plus goalkeeper in-situ task and b) explore any interactions with skill level. | 22 football players (defenders and/or defensive midfielders) were separated into two groups of 11 based on their tactical performance. | • 12 game sequences (2 vs. 1 + GK) were presented under high- and low-workload conditions. At the end of each sequence, participants indicated their perceived exertion.<br>• Visual search behaviours were recorded using Tobii Pro eye-movement registration glasses. | • Football players employ different gaze behaviours during low- and high-workload conditions.<br>• Physiological workload and tactical expertise interact to drive visual search behaviours within in-situ game situations. |
| Caso et al., [47] | To examine whether the amount of visual exploratory activity (VEA) is differentiated within a group of elite soccer players who have been assigned specific positions in the team. | Pre-recorded video-footage of 72 players from a Dutch elite soccer club was utilised. This included 29 first team players and 32 U23 players. Each team had players proportionately distributed across positions. | • VEA with respect to amount and timing for different player positions was assessed. | • Compared to players positioned in the defence and attack, midfielders showed the highest engagement in VEA.<br>• The amount of VEA during the penultimate pass predicted the success of subsequent passing actions.<br>• The amount of late VEA, during the final pass, did not significantly add to this prediction. |
| Coutinho et al., [89] | To examine how the manipulation of contextual dependency and decision-making (DM) affects players' positioning and subsequent ball control, passing performance, and external load during small-sided games (SSG). | 20 youth male football players from a Portuguese club competing at the regional level. | • Players performed one SSG followed by one of three experimental intervention tasks: (i) low DM (ii) moderate DM and (iii) high DM. The players then performed one additional SSG bout under the same conditions. | • More prescriptive tasks contributed to a higher frequency of actions but to a lower transfer to the subsequent game performance.<br>• Adopting tasks such as the game appeared to emphasise the coupling between perception and action, thereby enhancing the players' acute response. |

*(Continued)*

| Authors | Study Aim | Study Sample | Method | Important Results/Findings |
|---|---|---|---|---|
| Dambroz et al., [65] | To compare the visual search strategies of young soccer players based on positional roles. | 17, U13 soccer players from a club in the Serie A of the Brazilian National Soccer League, grouped according to their positional roles: defenders (n = 6); midfielders (n = 6); and forwards (n = 5). | • 11 video scenes of offensive soccer actions, recorded and watched through a third-person perspective. Mobile eye tracking was utilised to measure visual search strategies. | • Significant differences were found in total fixations in two locations: ball and teammates.<br>• Forwards focused more frequently on the ball and teammates when compared to midfielders.<br>• Forwards also fixated for longer periods on the ball when compared to midfielders. |
| de Joode et al., [145] | To establish within-group differences for game insight in elite youth soccer players; thereby, validation of the game insight indicator (GID) could be improved; (2) prospective value of perceptual–cognitive skills is examined by following the trajectory of the participants. | 14 youth soccer players playing in the U12 or U13 team for an elite youth academy in the Netherlands. | • The GID consisted of film clips that show game situations. Players had to predict the trajectory and destination of the ball and move toward the correct position to receive the pass of a teammate. | • GID is a useful indicator for anticipating the ball's trajectory and destination at age 11–13.<br>• No clear relationship with future performance was found.<br>• The GID can be a valuable tool to measure and evaluate perceptual–cognitive skills to examine the process of talent development. |
| Eldridge et al., [129] | To understand how experienced coaches working within a professional football academy conceptualise visual exploratory activity (VEA), and how they apply that conceptualisation in their design of practice activities. | 9 male football coaches working within an English Football Association Academy in the top four leagues of professional football in England. | • Semi-structured interviews were conducted to gain a deeper insight into each coach's perspective of VEA. | • Coaches hold VEA as an integral part of player performance which should be coached from a young age.<br>• Many coaches did not feel it necessary to dedicate the main focus within sessions to VEA.<br>• Within sessions, coaches recommended delivering VEA primarily through the design of practice activities, supported by visual aids (e.g., head- bands), video review, and questioning.<br>• Coaches desired additional evidence in support of various visual aids. |
| Farahani et al., [83] | To investigate the differences between Under 16, 18 and 23 academy soccer players on a video-based decision-making assessment and to compare participants' performance on the task in terms of accuracy and response time with their performance on the pitch. | 73 male participants. U23 (n = 26), U18 (n = 24), U16 (n = 23) academy footballers. | • Participants watched short video clips of soccer matches – 20 trials.<br>• Participants had to select best decision when video paused in time constrained conditions.<br>• Coaches were asked to rate their players 1–10 for decision-making in attacking and defensive scenarios. | • Response time was significantly lower in U23 compared to U18 and U16 and there was no significant difference between the U16 and the U18 groups, but no significant difference between age groups on the accuracy of response.<br>• Positive correlation between accuracy on the task and general decision-making skills rated by the coaches.<br>• Coaches' ratings of decision-making skills and response times on the task did not correlate. |
| Feist et al., [59] | To describe the visual exploratory activity of elite female central midfield players and understand the relationships between VEA, performance with the ball and specific contextual and situational factors in elite women's soccer. | 30 female central midfield players from the eight teams who competed in the knock-out stages of UEFA Women's European Championship 2022. | • Television broadcast and UEFA tactical footage were combined to analyse players across the seven knock-out stage matches, totalling 1,038 individual ball possessions. | • Pitch location was a significant predictor of scan frequency.<br>• Scan frequency was found to be a significant predictor for performance with the ball variables.<br>• Higher scan frequencies resulted in an increased likelihood of performing a successful action with the ball and performing a turn with the ball. |

*(Continued)*

| Authors | Study Aim | Study Sample | Method | Important Results/Findings |
|---|---|---|---|---|
| Fransen et al., [52] | To examine the influence of restricted visual feedback on the dribbling performance of youth soccer players. | 189 youth soccer players aged between 10–18 years were recruited from two Belgian high level football teams. Participants were divided into three groups of fast (n = 64, average (n = 63), and slow (n = 62) dribblers. | • Participants performed three assessment trials of a dribble test under different visual feedback conditions in a randomised order. | • Dribbling performance was significantly impaired with decreasing visual feedback.<br>• Decrements in dribble test performance with limited visual feedback were greater for fast dribblers when compared with slow dribblers. |
| Hicheur et al., [102] | To investigate if augmented feedback training can improve both perceptual–cognitive and motor skills during a passing situation in soccer. | 27 elite young soccer players – males, U14 and U15 categories. They belonged to the same elite youth soccer academy and were playing at Swiss-national level. | • Three groups of players performed a test consisting in passing the ball as accurately and as quickly as possible toward a visual target moving briefly across a large screen.<br>• The control group performed normal soccer training and was compared with two visuomotor training groups (augmented feedback and no-feedback). | • Only players of the augmented feedback group significantly improved passing accuracy, reactiveness, and global passing performance, whereas the non-feedback group only improved passing accuracy. None of these parameters was improved in the control group. |
| Hintermann et al., [49] | To investigate scanning in U19 female football players during 4v4 small-sided games (SSGs). | 61 female football U19 players from four teams were categorised into elite players (n = 31) and grassroots players (n = 30). | • 2010 game situations were video recorded for subsequent manual tagging. | • Elite players showed significantly more scans before, but not during ball possession than their grassroots counterparts in 4v4 SSGs.<br>• Number of scans before and during ball possession were significantly related to a players' success in the subsequent action, independent of the competition level. |
| Idarraga & Valencia-Sánchez, [104] | To assess the impact of psychokinetic games on tactical creativity, passing effectiveness, and ball control in youth women's soccer players. | 10 female U14 Colombian youth soccer players. | • An eight-week training programme with one psycho-kinetics games session per week lasting approximately 22 min, totalling 174 min. | • Eight sessions were sufficient to enhance tactical creativity with a moderate effect size. |
| Jordet et al., [41] | To investigate how professional soccer players use visual scanning in real games; establish the extent to which scanning varies under different contextual conditions; and to examine the relationships between scanning and performance. | 27 professional male soccer players aged 17–32 years. All players represented the same team in the EPL in the 2017/2018 season. | • Players were filmed across 21 matches, producing a total number of 9,574 individual ball possessions for analysis. | • Central midfielders and central defenders scan most frequently.<br>• Less scanning is done in tight scenarios, in wide areas and closer they get to the opponent's goal.<br>• Players scan more when they are going to pass as the next move, in particular long forward passes.<br>• The more a player scans, the higher probability of a successful outcome. |
| Keller et al., [84] | To determine if a video-based decision-making task could classify Australian older youth soccer players into different cohorts (levels of expertise) based upon their decision-making skill. | 62 male youth soccer players from the Western Australian U18 state league (n = 22), WA National Training Centre (n = 22) and the Australian Institute of Sport soccer programme (n = 20) and are referred to as sub-elite, state elite and national elite, respectively. | • 31 clips were randomly presented for 15 s in duration and then frozen for 5 s before being occluded.<br>• Participants were required to circle on a printout of the final screenshot, the most appropriate option to pass to, or circle the goal to indicate that they would shoot at goal. | • Video-based decision-making test is able to discriminate between differing levels of expertise for youth athletes. |

*(Continued)*

| Authors | Study Aim | Study Sample | Method | Important Results/Findings |
|---|---|---|---|---|
| Kent et al., [103] | To deliver and evaluate a theoretically informed performance under pressure intervention. | 82 male academy soccer players. | • Participants completed baseline pressure task.<br>• Players were randomly allocated to an intervention group; receiving pressure training (PT), three cognitive behaviour workshops, and reflective diaries or comparison group; receiving PT only.<br>• 68 players repeated the PT task at a six-week follow up (B), and of these, 26 (n = 15; intervention group; n = 11; PT only) also completed a re-test PT task (A) at 12-week follow up. | • Intervention players scored significantly higher in their decision-making with a significant main effect of age-group on decision-making and skill execution. |
| Leso et al., [127] | To identify the perception of creativity and game intelligence of soccer coaches and young players. | All athletes and soccer coaches from the 11-a-side Academic Association of Coimbra soccer team. 152 participated (118 players, 34 coaches). | • Questionnaire was used to investigate the perception of creativity and game intelligence of coaches and players. | • There were significant differences in the importance attached to creativity and game intelligence for the players.<br>• Coaches mostly associate the creativity in soccer to a kind of magical thinking, also adding the game intelligence to the ability of rationality, problem solving, and decision-making.<br>• There was a strong correlation between creativity and game intelligence for the players. |
| Li et al., [62] | To examine both the quantity and quality of players' memory and guess in pattern recall for complex soccer scenarios. | 72 participants, 36 soccer athletes (20 males, 16 females) and 36 non-athletes (14 males, 22 females). | • An adapted pattern recall task was used, in which participants were required to reproduce player locations in complex scenarios through two stages, allowing for the separation of memory from guesses and the subsequent analysis of accurate memory and false memory. | • Compared with non-athletes, soccer athletes maintained a greater number of players in accurate memory, while non-athletes falsely memorised more attacking players and tended to guess more defending players. |
| Machado, et al., [67] | To assess the visual search strategy of soccer players from different age groups. | 51 male youth soccer players from a Brazilian Serie A club. Players were equally grouped (n = 17) in age brackets U13, U15 and U17. | • 11 soccer offensive video sequences, through a third person perspective from European national championship games.<br>• Each video sequence was paused where participants had to answer "What should the player with ball possession do?". Eye tracking glasses were used to measure visual exploratory activity. | • Visual search strategy related to the number of fixations were able to discriminate U13 players from the U15 and U17 age groups.<br>• The U15 and U17 players did not present significant differences in their visual search strategy for both number of fixation and time of fixation. |
| Machado et al., [85] | To compare decision-making skills between selected and deselected players from elite Brazilian youth soccer academies. | 317 Brazilian youth elite male soccer players recruited from four soccer teams that played at the first national division in Brazil (i.e., U14, U15, U16, and U17). | • Players' decision-making skills, both quality and response time, were assessed with an objective video-based test (TacticUP®). | • Advantages in decision-making quality for U14 and U15 selected soccer players<br>• Advantages in decision-making time for U14, U16, and U17 selected soccer players compared to deselected players.<br>• Advantages of decision-making time for U15 selected players compared with those deselected. |

*(Continued)*

| Authors | Study Aim | Study Sample | Method | Important Results/Findings |
|---|---|---|---|---|
| Machado et al., [90] | To examine the relationship between the engagement in previous developmental activities in soccer and futsal with the quality and speed of decision-making skills in different phases of sport development of elite female soccer players. | 77 elite Brazilian professional female soccer players from six soccer teams of three different states in Brazil, competing in Brazil's professional national league. | • Players' decision-making skills were assessed based on video-based test – TacticUP®.<br>• Retrospective questionnaire was used to collect information on participation in different developmental activities. | • Engagement in deliberate practice in soccer and futsal, especially during childhood and early adolescence, is related to a better quality of offensive decision-making skills.<br>• Engaging in deliberate play in soccer, mainly in childhood and early adolescence, is related to quicker offensive and defensive decision-making skills of elite female soccer players. |
| Machado et al., [87] | To verify the influence of 25 training sessions based on tactical principles and small-sided and conditioned games (SSCG) in developing cognitive and motor decision-making skills of U12 soccer players. | 25, U12 elite male soccer players from a Brazilian first division national club. | • Perceptual-cognitive decision-making skills were assessed with an objective video-based test (TacticUP®).<br>• Perceptual-motor decision-making skills (tactical efficiency) were assessed with the System of Tactical Assessment in Soccer (FUT-SAT).<br>• 25 training sessions were organised based on tactical principles and SSCG. The sessions were designed considering the individual needs to improve decision-making skills. | • Improvements in perceptual-cognitive decision-making time for both offensive and defensive actions performed with the ball and near the ball and defensive actions performed far from the ball.<br>• Improvements in perceptual-motor skills (tactical efficiency) for defensive actions performed near the ball.<br>• Decision-making time develops quicker than the quality of decision-making. |
| Maskell et al., [130] | To investigate and interpret coaches' perspective of decision-making and visual exploratory activity (VEA) and what motivates their personal approach to developing VEA and decision-making. | 12 male coaches aged between 22 and 49 years old who were all employed by the same UK-based League One football club's youth academy. | • Individual, open ended, retrospective interviews. | • Coaches and players need to work together to create a shared understanding and common language using a combination of questions, phrases and video analysis to improve decision-making and VEA development.<br>• Integrating VEA and decision-making successfully in developmental sessions were viewed as most effective in opposed, chaotic training,<br>• Developing and delivering "chaotic" sessions was impacted by coach experience and development of the coach–player relationship. |
| McGuckian et al., [42] | To further understanding of how exploratory head movements relate to performance with the ball, and how this relationship changes with exploration in various time-periods before gaining possession of the ball. | 32 male outfield soccer players aged 16–30 years. Participants were recruited from clubs playing in the Australian National Premier League. | • Participants competed in 11 vs. 11 match-play.<br>• Outfield players wore a 9-degrees-of-freedom inertial measurement unit (IMU). | • Strong relationship between head turn frequency and head turn excursion.<br>• A higher-than-average head turn frequency and head turn excursion before receiving the ball resulted in a higher likelihood of turning with the ball, playing a pass in the attacking direction, and playing a pass to an area that is opposite to which it was received from.<br>• When players explored their environment with higher-than-average head turn frequency and excursion, they used more complex action opportunities afforded by the surrounding environment. |

*(Continued)*

**Table 3.** (Continued)

| Authors | Study Aim | Study Sample | Method | Important Results/Findings |
|---|---|---|---|---|
| McGuckian et al., [50] | 1) To gain a better understanding of the exploratory actions used by footballers in a perception-action football receiving-passing task; 2) to empirically test the relationship between head movements, as a proxy for visual exploratory action, and performatory action. | 12 male football players including 2 wide defenders, 4 defensive/central midfielders, 3 wide midfielders, and 3 attacking midfielders/strikers. Participants played for the same semi-professional club, competing in the Australian National Premier League. | • Participants completed a simulated receiving–passing task that required them to indicate pass direction to one of four surrounding targets, as quickly as they could after they gained simulated ball possession.<br>• The frequency of head movements before and after gaining ball possession and the pass response times were recorded. | • Time constraints of the task influenced the head movements and performatory actions of footballers in the passing task.<br>• The relationship between head movements and the speed of a passing response gives further evidence for the importance of exploratory action in service of the prospective regulation of movement. |
| McGuckian et al., [44] | To compare the visual exploratory actions of youth soccer players based on; pitch location, playing role, and ball possession phase. | 22 male outfield soccer players aged 15–17 years who had 8–13 years playing experience. Participants played for the same semi-professional club in the Australian National Premier League competition. | • Soccer players played a total of 1,623 minutes. Inertial measurement units, global positioning system units and notational analysis were used to quantify relevant variables. | • Players explored more extensively when they were in possession of the ball, and less extensively during transition phases, as compared to both team ball-possession and opposition ball-possession phases of play.<br>• Players explored most extensively when in the back third of the pitch, and least when they were in the middle third of the pitch. |
| Meira et al., [151] | To explore visual strategies employed by female soccer players of varying expertise levels. | 30 Brazilian female volunteers: 10 professional soccer players from a leading team of the Country's Major League, 10 varsity soccer players, and 10 novices. | • Participants were required to respond physically to film images projected on a screen using soccer skills (penalty kick, dribbling, pass reception, and defensive cover) whilst eye tracking technology examined gaze behaviour. | • Female elite players engaged in more visual fixations, directed gaze toward more relevant areas of the scene, and showed less variability of pupil diameter, compared to their non-elite counterparts. |
| Musculus et al., [86] | To test how environmental constraints affect decision-making processes in expert and non-expert players. | 80 male soccer players consisting of 40 experts (5th tier league in Germany) and 40 non-experts (two lowest German leagues). | • Video based decision making tool was used to present videos with manipulated opponent pressure, time constraint and first-person perspective. Players had to make a decision at moment of video occlusion. | • Players' option and decision quality improved under the time constraint but were negatively affected by opponent pressure.<br>• The negative effects of opponent pressure were especially true under limited time and in third-person perspective. |
| Natsuhara et al., [68] | To analyse eye movement data and verbal reports while soccer players one-touch passed the ball in a five-on-four attacking scene. | 18 high-level soccer players (HLP) and 18 middle-level soccer players (MLP). | • 5 vs. 4 four soccer-specific film simulations from previously recorded real scenes, where participants reacted to the life-sized soccer scenes. | • HLP group had significantly higher accuracy and consistency in decision-making compared with the MLP group.<br>• HLPs had a significantly greater number of fixation locations than MLPs. |
| O'Connor et al., [133] | To investigate the pedagogical approaches coaches use to develop decision-making in soccer. | 29 coaches currently working with youth soccer teams (U12, n = 5; U13, n = 8; U14, n = 4; U15, n = 4; U17, n = 8). All teams competed in the Australian National Premier League News South Wales Youth competition. | • Youth soccer coaches were filmed conducting two practice sessions.<br>• The first session was a regular training session, whereas in the second session, participants were asked to create an activity they believed would promote the development of on-ball decision-making. | • Strategies for incorporating decision-making into training activity – repetition of real scenarios with guided discovery, prompting decision-making by providing cues or solutions, and manipulating the game/activity.<br>• Coaches generally use strategies to develop decision-making but generally all coaches over coach with too much stop start and instructions. |

*(Continued)*

| Authors | Study Aim | Study Sample | Method | Important Results/Findings |
|---|---|---|---|---|
| O'Connor et al., [128] | To understand how coaches conceptualise decision-making and its development in soccer. | 25 participants (24 = male; 1 = female) who had a minimum of 20 years of experience in soccer. All participants had played professional soccer in Europe and Australia for an average of 14.8 years. Since retiring from playing, all participants had become coaches. | • Semi-structured interviews were carried out on participants to understand their thoughts, feelings and understanding of decision-making practices in soccer. | • Participants' conceptions of good decision-makers reflected the broadening range of abilities required of players.<br>• Participants highlighted their conceptions of how decision-making may be developed, emphasising the importance of playing with others; effective communication; balancing structure and autonomy; knowledgeable inspiration from other players and coaches; and a focus on improvement rather than winning. |
| Phatak & Gruber, [43] | To examine the relationship between visual exploratory frequency (VEF) and individual performance parameters in elite football midfielders. | Videos from 51 games of Euro 2016 football championships, were used where 35 male centre midfielders were used. | • Video footage analysed for player scans, transition scans, and total scans. | • Scanning rate of a player was positively correlated with his pass completion rate.<br>• Scan rate while receiving the ball was negatively correlated with turnover rate. |
| Práxedes et al., [88] | To analyse the effect of two training programmes, each based on modified games with different level of opposition, on decision-making and technical execution in two groups of young football players of different abilities. | 19 football players from the under-12 category of two teams from the same Spanish club consisting of 10 average skill and 9 low skill players. | • Participants experienced 4 phases; Pre-intervention 1, Intervention 1 (teaching programme based on modified games with numerical superiority in attack), Pre-intervention 2 and Intervention 2 (teaching programme based on modified games with numerical equality).<br>• Each intervention phase lasted 14 sessions. Decision-making and the execution of pass action during league matches over the same period were evaluated using the Game Performance Evaluation Tool (GPET). | • Average skill-level group showed significant differences after the first intervention in decision-making and execution of the pass action but not after the second intervention.<br>• For the Low skill-level group, significant differences were only observed in the execution of passing between the first and last phases. |
| Pokolm et al., [46] | To model football players' scanning, integrating players' pass completion rate, and their body orientation, in relation to individual and contextual variables such as a player's appearances in national teams and the opponent pressure the player is exposed to. | 8 selected matches and 11 different teams from the UEFA U17 and U19 European Championship in 2018, and 9 matches, including 12 different teams from the UEFA U17 and U21 European Championship in 2019. | • Video based data analysis of the games sampled. | • A player's appearances in national teams were not clearly related to body orientation and the pass completion rate.<br>• Players who played more matches for their national team performed more scans.<br>• Scanning had an impact on body orientation as well as on the pass completion rate.<br>• Opponent pressure had the most substantial effect on body orientation and pass completion. |
| Pulling et al., [134] | To further the understanding of coaches on visual exploratory activity (VEA) and coaching practices. | 303 current soccer coaches ranging in qualifications and experience. | • Participants completed an online survey comprised of 3 sections and 12 items. | • Those identified to have a high delivery of VEA were likely to provide more feedback/instruction on VEA; they designed an activity or part of a session to focus on VEA more often; and the percentage of sessions they would primarily focus on VEA was higher.<br>• Higher coaching qualification and experience leads to a positive attitude of coaching VEA. |

*(Continued)*

**Table 3.** (Continued)

| Authors | Study Aim | Study Sample | Method | Important Results/Findings |
|---|---|---|---|---|
| Roca et al., [60] | To examine anticipation and decision-making in skilled and less skilled soccer players. | 24 adult male soccer players were recruited. Skilled participants (n = 12) were professional (n = 6) and semi-professional (n = 6) players. Less skilled players (n = 12) had participated only at an amateur or recreational level. | • Skilled and less skilled players interacted as defenders with life-size film sequences of 11 versus 11 soccer situations.<br>• Participants were presented with task conditions in which the ball was in the offensive or defensive half of the pitch. | • Skilled players reported more accurate anticipation and decision-making than less skilled players, with their superior performance being underpinned by differences in task-specific search behaviours and thought processes.<br>• Visual search behaviour of skilled players involved more fixations of shorter duration to significantly more locations in the visual display. |
| Roca et al., [64] | To examine creativity in the decision making and visual search behaviours of skilled soccer players during simulated 11-a-side match play. | 44 skilled, male outfield soccer players. Players were recruited from a range of different professional and semi-professional soccer clubs in England. | • Players were required to interact with a representative life-size video-based simulation of attacking situations whilst in possession of the ball.<br>• Clips were occluded at a key moment, and they were required to play the ball in response to each situation.<br>• Moreover, they were required to name other additional actions they could execute. | • Creativity-based differences in judgment were underpinned by differences in visual search strategy.<br>• Most-creative players employed a broader attentional focus including more fixations of shorter duration and towards more informative locations of the display compared with least-creative players.<br>• Most-creative players detected team-mates in threatening positions earlier in the attacking play. |
| Roca & Ford, [131] | To investigate the types of practice activities used by coaches of elite youth players across European countries. | 53 male soccer coaches working with boys under 12 to under-16 age groups within 16 youth academy professional top-division clubs of four European nations. | • 83 practice sessions from under-12 to under-16 age groups were assessed – around 20 from each country for four clubs. | • More time was spent in active decision-making compared to non-active decision-making activities and transitioning between activities.<br>• Portugal and Spain spent a higher amount of time in active decision-making activities compared to English and German players.<br>• English players spent more time in unopposed technical-based drills and German players in improving fitness aspects of the game without the ball. |
| Roca, Ford & Memmert, [40] | To determine the underlying perceptual and cognitive processes that underpin creative expert performance in the sport of soccer. | 40 male outfield soccer players. These players were recruited from a range of different professional and semi-professional soccer clubs in Southeast England. | • Participants interacted with representative video-based 11 vs. 11 attacking situations whilst in possession of a ball. Clips were occluded at a key moment and participants were required to play the ball in response to each presented scenario.<br>• Moreover, they were required to name other additional actions they could execute for each situation. | • High-creative players made more fixations of shorter duration in a different sequential order and to more task-relevant locations of the display.<br>• High creative players generated a greater number of verbal reports of thoughts related to the assessment of the current task situation and planning of future decisions when compared with the low-creative players. |
| Savelsbergh et al., [54] | To examine whether differences in visual search and locomotion behaviour can be identified in a homogenous group of young soccer players who were selected as talented soccer players. | 20 male players, age 10–13 years. All participants played for the regional selection team of the Royal Dutch Soccer Association or for a professional club. | • Participants watched video clips of a 4 vs. 4 position game.<br>• Participants were asked to take part in the game by choosing the best position for the reception of the ball passed. | • The high-score group looked more to the ball area, while the players in the low-score group concentrated on the receiving player and on the hips/upper-body region of the passing player.<br>• The high-score group was better in moving to the correct direction and to the correct area to receive the ball than the low-score group. |

*(Continued)*

| Authors | Study Aim | Study Sample | Method | Important Results/Findings |
|---|---|---|---|---|
| Shui et al., [63] | To provide a measurement method of perceptual-cognitive skills in soccer. | 21 adolescent and 27 adult female soccer players were recruited from two provincial teams. | • The pattern anticipation task assessed participants' ability to anticipate player field locations from briefly presented soccer match video segments. | • All age groups were more accurate in anticipating the soccer ball and offensive players during counter attacks than during positional attacks.<br>• Anticipation accuracy for the soccer ball was lower than for the players, suggesting that the dynamic movement of the soccer ball is indeed challenging to anticipate. |
| Teoldo et al., [81] | 1) To compare the number of decisions made by soccer players at academy and professional levels in official matches; 2) To compare the quality and speed of decision-making between academy and professional soccer players. | Study 1–24 soccer teams, from which 14 were professional teams that played in the 2012/2013 UEFA Champions League, whereas 10 were U-20 academy teams that played national tournaments in Brazil.<br>Study 2–138 male soccer players, from which 42 were professional and 96 were young academy players (62 U17 and 34 U20 players). | • Study 1 – Notational analysis of 12 official matches.<br>• Study 2- TacticUP® was used to assess game reading and decision- making skills. | • Professional players made more decisions in official matches when compared to academy players.<br>• Professional players made quicker decisions than academy players in offensive and defensive actions, inside and outside the centre of play, and both with and without the ball. |
| Vaeyens et al., [27] | To examine differences in decision-making skill and visual search strategies across five categories of small-sided, offensive game simulations in soccer. | 87 male adolescents. The elite group consisted of 21 youth players. The sub-elite group included 21. The third subgroup consisted of 23 soccer players. | • Film simulations of offensive play, utilising movement-based response measures, and an eye movement registration. | • Playing experience, skill level, and the unique constraints of the task, expressed by the number of players and relative proportion of offensive and defensive players, determined both the observed search behaviour and processing requirements imposed on players in dynamic offensive team simulations. |
| van Maarseveen et al., [80] | To examine how well performance in a complex time-constrained motor task could be predicted using representative tests of perceptual-cognitive skill. | 22 highly talented female soccer players from the national soccer talent team participated in the study. | • 3 vs. 3 performance analysed by coaches before participants completed lab based, video interactions of 3 vs. 3 recordings. | • On-field performance could not be predicted on the basis of performance on the perceptual-cognitive tests.<br>• There were no strong correlations between the level of performance on the different tests.<br>• Perceptual-cognitive tests may not be as strong determinants of actual performance as may have previously been assumed. |
| Vater et al., [29] | To investigate perception-action coupling of defenders in 3 vs. 3 defensive situations and combine an on-field decision making test with verbal reports to examine how expert soccer players use their peripheral vision. | 10 high-skilled and 10 low skilled male soccer players. | • Mixed-methods approach with in-situ decision making (3 vs. 3 situations) and retrospective verbal reports to identify perceptual strategies used for optimal information pick-up in high- and low-skilled soccer players. | • High skilled players seem to actively monitor players' actions with peripheral vision, while low skilled players rely more on bottom-up information changes that are perceived with peripheral vision.<br>• The use of peripheral vision by central defenders depends on the position of the ball and the position of the direct opponent. |
| Vítor de Assis et al., [61] | To compare the decision-making and visual search strategies of young soccer players between two groups based on the results on-field specific tactical test. | 90, U15 male soccer players affiliated to three regional Brazilian clubs. | • Tactical behaviour was assessed using the FUT-SAT assessment tool.<br>• Video-based tests were used to assess decision-making, while visual search strategies were performed. | • More skilled participants showed better decision-making scores compared to the less skilled.<br>• The more skilled performed a higher mean number of fixations per trial and had a lower mean duration of fixation. |

*(Continued)*

**Table 3.** (Continued)

| Authors | Study Aim | Study Sample | Method | Important Results/Findings |
|---------|-----------|--------------|--------|----------------------------|
| Vítor de Assis et al., [28] | To compare the visual search strategy and anticipation between two groups of young players of different efficiencies in tactical behaviour. | 44 Brazilian male soccer players from three regional clubs. | • Tactical behaviour was assessed using FUT-SAT; anticipation score was obtained by a video-based assessment, while visual search strategy was performed using mobile eye tracking. | • There seems to be an interaction between cognitive skills such as anticipation and visual search strategy with the application of tactical behaviour by soccer players. <br> • The more tactically efficient soccer players showed a different visual search strategy and better anticipation judgments. |
| Ward & Williams, [33] | To examine how visual, perceptual, and cognitive skills develop as a function of age and skill in soccer. | Elite and sub-elite male soccer players (n = 137) were selected as participants. Elite players were recruited from English Premier League Academies, while sub-elite players were recruited from local elementary and secondary schools. | • Multidimensional battery of tests utilised to measure static visual acuity; dynamic visual acuity; stereoscopic depth sensitivity; and peripheral awareness. | • The tests of visual function did not consistently discriminate between skill groups at any age. <br> • Tests of anticipatory performance and use of situational probabilities were the best in discriminating across skill groups. <br> • Memory recall of structured patterns of play was most predictive of age. <br> • As early as age 9, elite soccer players demonstrated superior perceptual and cognitive skills when compared to their sub-elite counterparts. |

## Visual search behaviours

**Scanning.** Visual search comprises of scan frequency, scan excursion, and fixation locations [40], influencing a player's ability to make informed, goal-directed decisions under pressure. 31 of the 55 studies reviewed examined visual search behaviour through varying terminology (gaze strategies, scanning, visual attention, exploration activity). Scanning (an active head movement away from the ball to gather information in preparation for engaging with the ball) [41] is an important component of perceptual-cognitive expertise and a players visual search strategy, with direct implications for TD in soccer. Scanning has typically been measured through observational methods [41–47], due to the simplicity, cost effectiveness, and ecological validity [48], providing insights into the frequency and locations of visual search behaviours of professional/elite performers whilst being able to link them more specifically to performance variable outcomes [36,49]. Visual search analysis from live competition shows relationships between scanning frequency and location prior to receiving the ball and positive outcomes such as forward movements, body orientation, successful and more impactful passes, and turns out of pressure [42,44–47,50–52]. Furthermore, contextual variables such as playing position, pitch location, possession status, pressure on player-in-possession, time remaining, as well as team situation (winning, losing, drawing), impact scanning behaviours [41,45]. This provides the truest representation of visual search behaviours because of the environment players are exercising such skills [42,44,53].

Given that decision-making speed and accuracy are fundamental to performance given the highly dynamic nature of soccer, scanning behaviours could serve as a critical marker for assessing and developing perceptual expertise in young players. Thus, it is crucial to further develop research within *in-situ* contexts. Adding further methodological sophistication to current approaches, such as coupling eye tracking methodologies with observational analysis, may prove worthwhile. As Aksum et al., [51] highlights, 11 vs. 11 *in-situ* games do not include many fixations (2.3%) when scanning, whilst there is a whole body of research highlighting the importance of fixation locations [27,54,55]. This discrepancy underscores the need for further exploration to ensure practitioners receive accurate guidance on the role and requirements of scanning behaviours in player assessment and development. To effectively integrate scanning into TD processes, standardised

**Table 4. Theme analysis overview.**

| Theme Identified | Sub-Theme | Study Key Words | | |
|---|---|---|---|---|
| *Perceptual-cognitive skills* | Visual search behaviours | • Anchor<br>• Exploration<br>• Exploratory activity<br>• Eye movements<br>• Gaze behaviour | • Gaze strategies<br>• Orientation<br>• Pivot<br>• Scans<br>• Scan frequency<br>• Scanning<br>• Vision | • Visual attention<br>• Visual exploratory activity<br>• visual exploratory frequency<br>• Visual function<br>• Visual perception<br>• Visual search<br>• Visual search strategy |
| *Perceptual-cognitive skills* | Cognitive capabilities | • Affordance<br>• Anticipation<br>• Attention<br>• Awareness<br>• Cognition<br>• Cognitive effort<br>• Decision-making | • Game intelligence<br>• Locomotion behaviour<br>• Option generation<br>• Pattern recall<br>• Perception<br>• Perception–action<br>• Perceptual cognitive expertise | • Situation awareness<br>• Situational probabilities<br>• Spatial<br>• Tactical awareness<br>• Tactical knowledge<br>• Take-the-first heuristic |
| *Performance* | | • Ball control<br>• Expertise<br>• Game reading<br>• Motor control<br>• Motor skill<br>• Performance | • Expertise and expert performance<br>• Perceptual-motor skills<br>• Passing percentage | • In situ performance<br>• Passing behaviour<br>• Passing performance<br>• Response time<br>• Sports performance<br>• Tactical performance<br>• Technical performance<br>• Turnovers |
| *Methodologies/ interventions* | | • Assessment<br>• Augmented feedback training<br>• Behaviour<br>• Constraints<br>• Ecological psychology<br>• Eye-tracking<br>• Eye movements<br>• Feedback<br>• Game insight indicator<br>• Match analysis<br>• Match play<br>• Occlusion<br>• Occlusion task<br>• Performance analysis | • Performance intervention<br>• Playing position<br>• Practice activities<br>• Precues<br>• Representative design | • Representative learning design<br>• Situational task constraints<br>• Stroboscopic vision<br>• Systematic observation<br>• Task constraints<br>• Task representative<br>• Verbal reports |
| *Methodologies/ interventions* | Sport context/ participants | • Adolescence<br>• Athletes<br>• Coaching<br>• Deliberate practice/ play and futsal<br>• Elite athletes<br>• Elite youth<br>• Development<br>• Female football | • Skill acquisition<br>• Small-sided games<br>• Soccer<br>• Sport<br>• Football<br>• Pedagogy<br>• Practice design<br>• Team sports | • Talent development<br>• Talent identification<br>• Talent identification and selection<br>• Training tasks<br>• Women<br>• Youth<br>• Youth Sports<br>• Youth elite soccer players |
| *Miscellaneous* | | • Coping<br>• Engagement<br>• Learning<br>• Mediating mechanisms | • Mental toughness<br>• Pass receive<br>• Player-role<br>• Resilience<br>• Skill<br>• Tactics | • Yo-Yo test |

assessment protocols should be established, capturing both quantitative (e.g., frequency, duration) and qualitative (e.g., information extraction) aspects of scanning. Developing such frameworks will enhance the accuracy and applicability of perceptual-cognitive assessments, ultimately supporting practitioners in refining evaluation and development processes.

Participants in observational studies examining scanning behaviours include players from the Men's and Women's European Championships [43,52], U17-U19 Men's European Championships [46], English Premier League (EPL; [41]), Eredivisie Dutch League [47], English (U17-U19; [45]), and Australian (U15-U17; [44]) boy's youth academies, Swiss elite and grassroots women's teams [49] as well as men's semi-professional teams [42]. This broad range of levels has provided a comprehensive perspective of scanning behaviours in match scenarios. While these studies contribute valuable insights into how PCS are exercised in high performance environments, there remains a lack of research on visual search behaviours in younger age groups. Existing studies primarily examined players already competing at elite or near-elite levels, but little is known about how scanning behaviours emerge and develop in younger players during key developmental phases. Research in youth academies has focused on older age groups (e.g., U15-U19, where players are transitioning into professional environments), yet limited data exist on players aged U9-U14, a critical period for PCS acquisition and long-term skill development [56]. Future research should extend current research samples to explore scanning in more diverse developmental contexts. This would provide a more holistic understanding of how visual search behaviours evolve across levels, enabling practitioners in TD environments to implement age-appropriate training and assessment strategies. By identifying how scanning abilities progress from early development to professional levels, practitioners can better support players in refining scanning behaviours, ultimately enhancing long-term development outcomes [41,57].

**Fixation behaviours.** Beyond scanning actions, fixation behaviours (i.e., what players are looking at) pre, concurrent, and post action-execution are crucial to understanding soccer player's visual search capabilities [58]. Research suggests that basic visual function (e.g., visual acuity) does not distinguish skill levels across youth age groups [33], reinforcing the need to investigate how players use their gaze strategically rather than just their visual 'hardware'. Advances in eye-tracking technology have enabled the precise measurement of fixation locations, durations, and gaze strategies across different in-game moments [59], providing valuable insights into how gaze strategies are utilised in players of varying abilities.

Representative task designs are commonly used to study fixation behaviours, incorporating key decision-making mechanisms under game representative conditions, whereby several key behavioural mechanisms have been determined [54,55,60,61]. Typically, these include temporal occlusion tasks, where players anticipate future events based on static or live images, responding via button presses [55,62,63], movement decisions [54], passing actions [27], or verbal reports [64]. In approaches utilising representative designs, time to first fixate on the ball and leg area have been shown to underpin successful anticipation and decision-making skills [54,55]. In offensive scenarios, fixating on the player-in-possession improves reaction times in low number overloaded attacking scenarios (2 vs. 1), yet when numbers increase (5 vs. 3) fixating on teammates who are closely marked improves reaction time and accuracy of decision [27]. This pattern continues in 11 vs. 11 scenarios, where players with high anticipation skills used significantly more fixations of shorter duration and toward greater number of locations [54,55]. Furthermore, highly creative players also employ a broader attentional focus towards more informative locations and in a different sequential order, whilst also detecting teammates in threatening positions earlier in the attacking scenario [40,64]. Overall, there is a strong understanding of key differences regarding visual search behaviours, where it appears that more fixations of shorter duration can improve anticipation and decision-making processes [28,65,66].

From the current understanding surrounding fixation behaviours, there are some direct implications for future TD research. Fixation behaviours appear to differentiate players based on position, age, skill level, and cognitive decision-making systems [28,65–69], while also being influenced by high physical workloads [70]. However, current research predominantly isolates visual search metrics, limiting our understanding of how scanning, fixation behaviours, and gaze strategies interact in real game situations. Progressing from the work of Aksum et al. [51], future research

should explore the integrated relationship between scanning and fixation behaviours, enabling practitioners in TD environments to better understand how these skills develop and how training interventions can optimise perceptual-cognitive abilities in young players [71].

**Peripheral vision.** Peripheral vision enables players to detect information in the visual field outside the limits of central/foveal vision (≈99% visual field; [72,73]). Given the dynamic nature of soccer, players cannot always rely on scanning behaviours to gather visual information [22,74]. Verbal reports (verbalised mental thoughts) following 3 vs. 3 scenarios indicates highly skilled soccer players actively monitor opposition and teammates actions via peripheral vision, suggesting its role in anticipation and decision-making [29]. While interest on the importance of developing peripheral vision in soccer is increasing [75,76] there is limited literature support to guide TD practitioners. The focus remains on what players can see and interpret within their direct field of view. There is a pressing need to explore how peripheral vision information processing can be developed within TD environments and how its role in can be enhanced through effective practice designs [76,77]. Future research should seek to integrate peripheral vision training into structured development programmes, ensuring that practitioners can assess and train these underdeveloped yet potentially crucial PCS [76].

### Cognitive capabilities

**Decision-making and anticipation.** Decision-making in soccer has traditionally been subdivided into accuracy and speed [78–80], with anticipatory abilities influencing speed [22]. Professional soccer players make faster and a greater number of decisions both in and out of possession [81]. At youth levels, players exhibit different decision-making speeds both between and within ages at academy level [82], but no differences in accuracy between the same ages [83], where those exhibiting higher technical skills make more accurate decisions within age groups [61] and is independent of anticipatory skills [60]. Video-based decision-making tasks have been effective in distinguishing levels of decision-making among youth athletes [84], where Machado et al., [85] suggests there are differences in decision-making quality and time for those selected and not selected in youth development systems, highlighting the importance of understanding decision-making behaviours. To deepen our understanding, better integrating environmental constraints (i.e., time, pressure) to examine the impact of decision-making capabilities is required to provide further context to our understanding [86], where small-sided games approaches have been recently trialled [87]. This, in turn, will support TD environments where coaches can develop aspects such as practice design and coaching interventions more effectively to improve decision-making development alongside the current understanding of differences in high versus low performing athletes [88–90].

Advancements in research are moving towards a more integrated approach, uncovering the interactions between various PCS components. Specifically, anticipation and decision-making efficiency are influenced by saccadic eye movements (ballistic movements between fixations) and visual search strategies [55,60]. Higher-skilled players tend to make use of more fixations of shorter duration, targeting more task-relevant locations in a different sequential order than lower-skilled players [27,64]. Although this knowledge is beneficial for TD purposes, understanding whether TD practitioners themselves recognise the integrated links between refined visual search behaviours, anticipation, and decision-making could help researchers assess how well current findings are being translated into real-world coaching contexts [37], allowing tailored research approaches based on practitioner knowledge.

**Perception-action coupling.** Pairing cognition with motor-execution in applied soccer environments is fundamental for coaches designing sessions to facilitate improvements in motor performance [37]. Research has primarily focused on decision-making and anticipation in isolation, helping practitioners understand the differences between higher and lower skilled athletes [55,60,82,83,91]. However, to better understand the link between perception and action, there is a need to connect stimulus recognition speed, accuracy, and decision-making speed, as well as motor-execution; the antecedents to the action are required to achieve a more accurate perception-action coupling appreciation [92]. More information of whether enhanced PCS improves speed, weight, and accuracy of the pass, change of direction, the distance a player

gains from their opposition when receiving a pass, or the number of touches required prior to a successful shot on goal are examples of the more in-depth information surrounding PCS execution that future research may need to explore to increase practitioner (i.e., coaches) buy-in, providing relevant and actionable insights.

Representative task designs have already demonstrated value in bridging the gap between laboratory findings and applied practice. However, a more comprehensive understanding of perception-action coupling within the applied soccer environment is necessary to fully optimise TD processes [93]. If coaches better understand the role of PCS in real-game scenarios, where players' skills are tested under game-specific demands, they will be better equipped to design training sessions that foster perceptual-cognitive development, leading to a more effective transfer of research into practice [37,94–96]. Moreover, retrospective verbal reports and 'think aloud' strategies [60,97] could be valuable tools for enhancing perception-action coupling in applied settings, offering insights into the cognitive processes behind motor execution and further supporting coach education in TD contexts [98].

## Performance

**Game applications.** Successful soccer performance is often associated with various metrics such as successful passes, goals scored, tackles won, distance covered etc. [99–101]. Research on visual scanning behaviours and the relationship with performance metrics in 11 vs. 11 game formats has made significant strides in supporting TD practitioners [45]. However, the concept of performance in soccer is multifaceted and has been explored in various contexts within the PCS domain. These include passing performance following augmented feedback [102], 2 vs. 2 performance after pressure training interventions [103], eye function performance [33], decision-making performance linked to visual search strategies across different game contexts [27,61], and creative performance mechanisms utilising representative task designs [40,64] or novel training approaches [104]. Understanding performance in competition is crucial for providing practitioners a lens on current player capabilities and remains an area that requires further exploration. There is a significant distinction between performance in test settings and performance in soccer competition. To enhance TD processes, it is important to bridge this gap by demonstrating the real-world impact of enhanced PCS on match outcomes. By clearly linking perceptual-cognitive development to on-field performance, practitioners will be better equipped to integrate research findings into practical training environments and optimise player development [105].

**Skill execution.** The *Expert Performance Approach* aims to understand how expert performers acquire/refine skills across different sports and other domains [106]. Sport specific PCS literature examined differences between high and low performers and the mechanisms that underpin these differences due to the acceptance of differences in motor capabilities, thus there are likely differences in their PCS [107–109]. Currently, there is no literature that has examined any key developmental milestones of PCS expertise. Therefore, it becomes difficult for practitioners to identify what mechanisms develop, at what rates and in which order that enable individuals to become experts in adulthood. Using a combination of both laboratory- and field-based approaches would be advantageous to gain a greater understanding of how soccer players PCS performance is impacted, and what could be done from an applied perspective to develop such mechanisms. As there is currently limited information on PCS development, TD practitioners face challenges in systematically tracking or intervening in these areas [17]. Without a clearer understanding of PCS progression, practitioners strategies may risk being built on assumptions rather than evidence-based frameworks [110]. It is also noteworthy that there remains an inconsistency in classifying a 'skilled' (particularly high and low) or 'elite' athlete, which will ultimately shape the conclusions made if only examining either ends of the performance spectrum [111].

## Methodologies: Research designs

**Task designs.** Representative task designs, such as video and image-based occlusion, combined with interactive methodologies, have enhanced our understanding of how to develop PCS knowledge in soccer [112]. These designs aim to simulate perception-action coupling within game-based scenarios, where with the support of high technology

equipment, has provided standardised and repeatable conditions; facilitating greater insights into the mechanisms that underpin successful PCS in soccer [113,114]. Such tasks sacrifice ecological validity in favour of more reliable data, remaining the dominant approach in the field. However, these designs often fail to replicate the dynamic and complex nature of *in-situ* tasks, such as those encountered in real match scenarios [37]. For example, most visual search fixation behaviour research has been conducted in laboratory settings, making it difficult (even with applied recommendations), for practitioners to assess its direct applicability to real-world soccer performance. Such tests are also lacking correlation between *in-situ* performance values provided by coaches across multiple 3 vs. 3 scenarios and measures of anticipation, decision-making, and pattern recall measured through typical task designs [80]. This highlights a gap in the research and suggests that more *in-situ* measures are needed (or upskilling of coaches), an area where studies remain relatively limited [41,42,44,45]. Developing *in-situ* assessments would significantly enhance the practical applicability of this research area, allowing TD practitioners to better understand how PCS function in real-game contexts. By bridging this gap, practitioners would be in a stronger position to design more effective training interventions and assessment tools for player development.

**Technological advancements.** Technology has the potential to bridge the science-to-application gap within TD contexts, offering practitioners a better balance between subjective and objective data [115]. For example, mobile eye tracking technology has enhanced our understanding of gaze and scanning strategies, allowing a distinguishment between high- and low-skilled athletes [116,117]. The primary aim is to utilise technological advancements to achieve a better understanding of what players experience, read and respond to during a game of soccer, where the intensity is higher and the players approach to the game is different due a variety of contributing factors (e.g., psychological; motivation, anxiety etc.) that will be interacting and exercised simultaneously [118–121]. As stated previously, Aksum et al., [51] provided the most significant integration of eye tracking technology within live settings to date, collecting data from four players using eye tracking glasses during 11 vs. 11 matches. This approach should be expanded to other contexts to further our understanding of fixation behaviours in players and improve the data quantity available for making more robust conclusions.

A further promising development is the use of virtual reality (VR) technology, where speed, decisions and actions are more representative than before whilst providing greater reliability and the feeling of immersion within a real-life experience [122,123]. Though not perfect due to extraneous variables and inconsistencies of contextual variables the game of soccer provides, it is a positive step for TD programmes to consider [124]. For example, Soccer Bot $360^0$ technology facilitates greater game representation, with 360-degree environments, multiple task formats and diverse stimuli [125]. However, it currently lacks sophistication without the users ability to influence the situation being presented in any way, decoupling perception and action [114]. As VR technology continues to evolve, there is a compelling reason to explore its applications within TD settings to determine if, and how, it can support practitioners. This integration will require careful consideration of the balance between reliability and validity in the measurements taken, as trade-offs between these factors will inevitably exist in any research or practical setting.

## Methodologies: Sporting context and participants

**Coaches.** Coaches play a central role in TD processes (alongside varying input on TI), as their knowledge, abilities, and behaviours have a significant impact on player development [126,127]. Although there is a developing body of research regarding players PCS which coaches can interpret and utilise, there is a lack of research with those directly responsible for identifying and developing PCS. Coaches' conceptualisations of decision-making, visual exploratory activity and creativity [128–131] have become clearer, alongside understanding coaching approaches in professional academies surrounding time spent in decision-making activities [14,132,133]. Although coaches appear to be spending more time in decision-making practices, they are not currently optimising coaching styles and interventions to elicit these abilities [91,129,134]. This gap is influenced by coaches' perceptions of PCS, their level of experience, and their

qualification status, all of which impact coaches engagement developing visual exploratory activities and subsequent decision-making capabilities during training sessions [130,131,135].

While research has made positive strides in understanding coaches' perceptions of specific PCS, it remains limited in scope. The identification, monitoring, and development of PCS in athletes are crucial for coaches, as they are the primary individuals who need to translate sport-specific knowledge into actionable practices [136,137]. Further exploration of how coaches' perceptions of PCS align with the coaching practices they currently deliver could help better link knowledge of PCS and applied practice. Investigating how coaches identify and assess PCS in athletes may help refine current practices and improve their application in real-world settings. If coaches are unable to identify PCS, they may struggle to develop them effectively [98,138].

**Scouts.**  Within TI contexts, the primary role of a soccer scout is to identify youth players with high potential, a task that is complex and challenging [139]. Traditional approaches to TI include reliance on physical attributes, and scouts must be able to overcome their unconscious biases to assess potential, incorporating a players PCS, which does not necessarily present as clear, observable behaviours [140,141]. Currently, there is limited understanding of how scouts specifically identify PCS, making it difficult to assess how effectively they incorporate these skills into their evaluations. Clarifying how scouts measure and identify these abilities could provide valuable insights for practitioners, enhancing their understanding of the factors that currently contribute to TI.

It is also important to examine whether scouts' evaluations are still influenced by factors like chance events (e.g., relative age effect [16]) or physical skills [142], or if increased awareness of performance predictors has shifted their priorities toward more holistic skill assessments [143]. Although we know which skills scouts and coaches typically prioritise in TI and the agreed importance of PCS [143,144], the practical translation of this knowledge into their assessment processes remains unclear [144]. Mobile eye tracking technology offers an opportunity to explore the visual search strategies that underpin scouts' decision-making, alongside verbal reports on their evaluations of players. These insights could help refine the TI process by shedding light on the factors influencing identification decisions. Furthermore, decision-making assessment tools are being developed to support scouts in their evaluations, but these tools still require further PCS refinement before full integration into TI settings [145,146].

**Players.**  PCS in soccer have been examined across a wide range of player contexts, including adult and youth athletes, male and female players, and individuals of varying skill levels, experience, and competition standards. A common approach has been to compare high-skilled and less-skilled players to identify key differentiating factors. However, the PCS of players throughout specific TD pathways is less clear. For example, previous studies examining PCS in high performing soccer players have recorded data from players within English Premier League academies (category one) to explore areas of PCS expertise [33,83,104]. However, whether there are differences in youth academies further down the TD pyramid in England (e.g., category two or three) would be beneficial to understand if PCS are a substantial differentiating factor of players in the top tier academies, or if there are differences in identification or developmental processes from clubs at the top of the developmental pyramid versus further down. Providing data or tools that can demonstrate any clear PCS differences (building from Bennett et al., [145]) may then allow for improved recommendations for key stakeholders (e.g., academy manager, head of coaching) who make decisions on club policies and practices regarding TD.

**Female soccer.**  Soccer continues to be one of the most popular female sports worldwide [147], with its popularity rising exponentially (i.e., live/tv attendance; professionalisation etc. [148]). Consequently, professional female soccer clubs and nations are advancing their TD processes. For example, in England, professional female clubs are now advancing their talent pathways similar to, and in line with, the men's pathways, with new structures providing up to 70 Emerging Talent Centres (ETC) allowing 95 per cent of players access to an ETC within one hour of where they live by 2024 and the number of young female players engaged in FA talent programmes across the country rising from 1,722 to more than 4,200 by the end of the 2023−24 season [149]. While the scientific literature on female soccer is gradually

expanding [150], much of the soccer-specific research on PCS and TD has focused predominantly on male players. Studies examining TI between 1999 and 2019 show that only 14% of these studies focused solely on female participants [151]. While the proportion of PCS focused studies involving female athletes has increased, especially from 2020 onward [49,52,63,104,152], the need for more female-specific research remains. Extrapolating findings from male soccer to female players may lead to erroneous conclusions [153], as was previously noted from a technical perspective [154]. Given that the technical components of female athletes may differ, it is crucial that the same level of research curiosity is given to the PCS of female athletes. Investigating these components in female athletes will be essential for developing tailored and effective TD practices.

## Limitations and future directions

This review is not without its limitations. Firstly, we only included English language, peer-reviewed studies. Excluding non-English studies may influence sample characteristics (e.g., location) and lead to the omission of potential correlates that may be of cultural significance [155]. Yet, some evidence suggests that limiting our search to studies published in English may not always be ample to impact review findings [156]. Furthermore, a general weakness of scoping reviews concerns the lack of quality assessment of the included studies [155].

Based on the findings of the scoping review, several key areas emerged for future research that will further the fields understanding of PCS in soccer, particularly within TD contexts. These recommendations are designed to address the current gaps in knowledge and refine the practical applications of PCS. Each recommendation outlines a specific focus that could improve the identification, assessment, and development of PCS for youth athletes, guiding future research efforts:

1. **Incorporate practitioners in TD contexts:** Research should examine the roles of coaches, scouts, and other practitioners in identifying, monitoring, and assessing PCS in youth soccer players. Understanding how these stakeholders conceptualise PCS and integrate them into their practices will provide valuable insights into the real-world application of research findings and facilitate the adoption of evidence-based practices within TD environments.

2. **Bridge the gap between representative task designs and real-world performance:** While current research has made valuable contributions through representative task designs, further research is needed to connect these controlled settings with real-life scenarios. Future studies should focus on examining PCS during match play, using field-based methodologies, to understand how skills such as anticipation, decision-making, and gaze strategies manifest in high-pressure, dynamic environments. This will help to ensure that research findings are more ecologically valid and relevant for applied contexts.

3. **Implement developmental monitoring of PCS in TD processes:** Longitudinal studies that track the development of PCS across different age groups and stages of player development are currently absent. Understanding how PCS evolve over time in response to training, competition, and maturation will allow for more tailored development programmes and more sophisticated identification capabilities. These studies should investigate how PCS impact performance trajectories, helping to identify athletes with high potential earlier in the development pathway.

4. **Examine the interaction between different PCS and their interdependencies:** Further research is needed to explore how various PCS (e.g., decision-making, anticipation, visual search, etc.) interact with one another within *in-situ* contexts. Investigating how deficits in one area may influence performance in other PCS domains could provide a more nuanced understanding of how to address weaknesses and optimise player development. This will also help to develop more comprehensive interventions that target multiple PCS in a coordinated manner.

5. **Expand research to include diverse player groups:** Much of the current research on PCS has focused on male players, but it is essential to explore how these skills manifest in the women's game and across all TD age categories and

levels. Research should aim to identify any gender-specific differences in PCS and understand how the developmental needs of female players differ from those of male players. Additionally, studies should examine players at various stages of their development, from youth to professional levels, to ensure that PCS training is adapted to the changing demands of players as they mature.

## Conclusion

This scoping review provides an original contribution to the growing body of research on PCS in soccer, offering a comprehensive overview of the current state of knowledge in soccer-specific contexts. By synthesising key findings across a range of studies, this review identified critical gaps in the literature and highlights the need for further research to advance the identification, monitoring and development of PCS in youth athletes. While there has been notable progress in understanding the role of PCS in differentiating high and low skilled, significant gaps remain, particularly in terms of how these skills can be effectively identified, assessed, and developed through applied practices. To move the field forward, future research should focus on better defining the underlying mechanisms of PCS and how these skills can be reliably measured in applied settings, as well as exploring how training interventions can improve these skills in youth players. Additionally, a more nuanced understanding of how TD practitioners perceive and integrate PCS into their practice is essential for bridging the gap between theory and practice. As PCS expertise plays a pivotal role in soccer performance, addressing these gaps will enhance development processes and strategies, ultimately improving the overall quality of players emerging from TD systems.

## Supporting information

**S1 Table. PRISMA-ScR checklist (Tricco et al., 2018).**
(DOCX)

## Author contributions

**Conceptualization:** Andrew O. Triggs, Matthew Andrew.

**Data curation:** Andrew O. Triggs.

**Formal analysis:** Andrew O. Triggs, Matthew Andrew.

**Methodology:** Andrew O. Triggs, Matthew Andrew.

**Project administration:** Andrew O. Triggs.

**Supervision:** Joe Causer, Allistair P. McRobert, Matthew Andrew.

**Writing – original draft:** Andrew O. Triggs, Joe Causer, Allistair P. McRobert, Matthew Andrew.

**Writing – review & editing:** Joe Causer, Allistair P. McRobert, Matthew Andrew.

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
