## [Decision Letter · Decision Letter 0]

Dear Dr. Triggs,

Thank you for submitting your manuscript to PLOS ONE. After careful consideration, we feel that it has merit but does not fully meet PLOS ONE’s publication criteria as it currently stands. Therefore, we invite you to submit a revised version of the manuscript that addresses the points raised during the review process.

**ACADEMIC EDITOR:**

We look forward to receiving your revised manuscript.

Kind regards,

Leonardo Vidal Andreato, PhD

Academic Editor

PLOS ONE

Journal requirements:   When submitting your revision, we need you to address these additional requirements. 1. Please ensure that your manuscript meets PLOS ONE's style requirements, including those for file naming. The PLOS ONE style templates can be found at https://journals.plos.org/plosone/s/file?id=wjVg/PLOSOne_formatting_sample_main_body.pdf and https://journals.plos.org/plosone/s/file?id=ba62/PLOSOne_formatting_sample_title_authors_affiliations.pdf. 2. We noticed you have some minor occurrence of overlapping text with the following previous publication(s), which needs to be addressed: https://www.sciencedirect.com/science/article/abs/pii/S1469029221000790?via%3Dihub https://findresearcher.sdu.dk/ws/files/170526896/Kuettel_Larsen_2019_Mental_health_scoping_review_IRSEP.pdf? In your revision ensure you cite all your sources (including your own works), and quote or rephrase any duplicated text outside the methods section. Further consideration is dependent on these concerns being addressed. 3. We note that your Data Availability Statement is currently as follows: [All relevant data are within the manuscript and its Supporting Information files.] Please confirm at this time whether or not your submission contains all raw data required to replicate the results of your study. Authors must share the “minimal data set” for their submission. PLOS defines the minimal data set to consist of the data required to replicate all study findings reported in the article, as well as related metadata and methods (https://journals.plos.org/plosone/s/data-availability#loc-minimal-data-set-definition). For example, authors should submit the following data: - The values behind the means, standard deviations and other measures reported;- The values used to build graphs;- The points extracted from images for analysis. Authors do not need to submit their entire data set if only a portion of the data was used in the reported study. If your submission does not contain these data, please either upload them as Supporting Information files or deposit them to a stable, public repository and provide us with the relevant URLs, DOIs, or accession numbers. For a list of recommended repositories, please see https://journals.plos.org/plosone/s/recommended-repositories. If there are ethical or legal restrictions on sharing a de-identified data set, please explain them in detail (e.g., data contain potentially sensitive information, data are owned by a third-party organization, etc.) and who has imposed them (e.g., an ethics committee). Please also provide contact information for a data access committee, ethics committee, or other institutional body to which data requests may be sent. If data are owned by a third party, please indicate how others may request data access.

Reviewers' comments:

Reviewer's Responses to Questions

**Comments to the Author**

1. Is the manuscript technically sound, and do the data support the conclusions?

Reviewer #1: Partly

2. Has the statistical analysis been performed appropriately and rigorously?

Reviewer #1: N/A

3. Have the authors made all data underlying the findings in their manuscript fully available?

Reviewer #1: Yes

4. Is the manuscript presented in an intelligible fashion and written in standard English?

Reviewer #1: Yes

Reviewer #1: Dear authorship team,

Thank you for the opportunity to review your ScR. For brevity I have added more specific comments on the original submission. My comments that require a response are illustrated by AR (Action Required). Below I have summarised my comments and how I have arrived at my overall decision.

1. There is a recurring theme that runs through the whole paper (conceptual clarity around language). There are numerous phrases, sentences, claims that use appropriate terminology but don't actually tell me anything. I have used the term 'vague' and/or 'superficial' throughout the manuscript. This needs to be rectified.

2. I find the absence of a clearly defined RQ problematic. I accept the premise that ScR's are broader in focus but they are defined by a clearly defined question. This is a deal-breaker for me. I also wonder whether the TID/TDE has muddied the waters here. The PCS in soccer seems (to me at least) to make sense and indeed reflects the studies you capture. The TID/TDE aspect is not so defined and reads like an 'add-on'. The search syntax for instance doesn't capture all key stakeholders in the TID/TDE context (i.e. scouts/players). The inclusion criteria section is a good example of what I mean here. This needs to be much clearer. What soccer environments (clubs, schools, colleges etc...). There is also some confusion here. I assume by 'outfield players' you don't include GK's, but you have included GK populations in your review? What study design are you accepting (I assume all??? Quants/Quals/Mixed Methods) but not clear.

3. The introduction needs to be re-framed. I would lose the opening preamble and get down to the PCS in TID/TDE. I would however ensure this section ends with a really defined RQ for the review. I would also like to read how this review differs from other PCS reviews in talent (Dennis Murr) or coaching (Fynn Bergamn). Which takes me the point about why this was a ScR and not a full SR? This seemed to be a strange decision. I am not suggesting you change this to a SR, but on the basis there are reviews that explore PCS in other contexts I would like to read a more robust rationale for choosing a ScR. This section is critical (deal-breaker) as it needs to frame the context of PCS and TID/TDE just providing a general overview of TID/TDE is not enough in this context. My reading at the moment is that you are examining two binary constructs and not the integration of the two. Greater clarity will be key here.

4. I would re-run the search strategy. It was run 4 months ago. There may be some additional studies to include.

5. Re-visit the data extraction and analysis strategy. More specific comments are on the manuscript. This was very vague and needs to be more refined.

**Do you want your identity to be public for this peer review?** For information about this choice, including consent withdrawal, please see our Privacy Policy

Reviewer #1: **Yes: ** Simon J Roberts

---

## [Author Response · Author response to Decision Letter 1]

20 Mar 2025

Dear Dr Andreato,

Thank you for the valuable feedback and insightful comments provided by the reviewer on our manuscript entitled “Perceptual-cognitive skills and talent identification in soccer: A scoping review of the last 20 years” (Manuscript ID: PONE-D-24-10341). We appreciate the time and effort the reviewer has taken to assess our work, and we are grateful for the opportunity to revise and improve our submission.

We have carefully considered and addressed all the comments and suggestions provided. Below, we outline the changes made in response to each comment made by the reviewer. We believe that these revisions have strengthened the manuscript significantly. Please find resubmitted our revised manuscript: “Perceptual-cognitive skills and talent development environments in soccer: A scoping review”

The resubmission has been approved by all authors, and if accepted, it will not be published elsewhere.

Reviewer #1

Comment 1: Re-word doesn't make sense (line 23-25).

Response: We thank the reviewer for highlighting this issue. In response, we have re-worded the abstract introduction. This change can be found in line 1-5.

Comment 2: Depends on how you define performance (line 23).

Response: We appreciate the reviewer’s feedback on this point. We maintain that perceptual-cognitive skills are a required component of performance within professional soccer and have re-phrased to capture this more suitably. This change can be found on line 1.

Comment 3: - used above. Be consistent (line 26)

Response: We appreciate the reviewer’s feedback on this point. We have now made sure perceptual-cognitive skills maintain the use of the hyphen whenever the term is written in full throughout the manuscript.

Comment 4: For richer clubs I would add big data and machine learning (line 41-44).

Response: We appreciate the reviewer’s feedback on this point. As suggested, we have aimed to capture the incorporation of data science within talent programmes. This addition can be found on line 21.  

Comment 5: This is too vague and superficial. Re-word and summarise what you think is the culmative evidence of the work above (line 52-54)

Response: We thank the reviewer for highlighting this issue. As suggested, we have re-framed this paragraph to highlight the breadth of research surrounding talent development environments with a lack of research to show it’s applied impact and the use of evidence-based principles/frameworks. The revised text is now located in line 28-33.

Comment 6: I would lose this. I don't think you need to go down this rabbit hole. I would get straight into the PCS debate (line 55-66).

Response: We agree with the reviewers comments and have removed this paragraph as a result, moving straight into the PCS component of the introduction.

Comment 7: In view of my earlier suggestion this needs to be 'beefed up' and more relevant to TID/TDE processes/outcomes. I also need to confirm at this stage the focus will be academy/professional settings. I am reading words such as 'expert', 'performance' etc.. (line 67-82).

Response: We appreciate the reviewer’s feedback on this point. We have modified this paragraph to provide a clearer overview of PCS impact and importance within soccer and how it can support them moving towards future expert performance. The link with TDE is hopefully now clearer, where we have highlighted the need for practitioners to identify, assess and develop these skills to support players with the required skills for first team football. Although expert and performance terms are still incorporated, this is only referring to how we know PCS support future expert performers, and this is why TDE practitioners should invest in supporting athletes with these skills. This change can be found in line 42-55.

Comment 9: This confirms my point above. This is not supported and too vague. (what does detailing the manner these skills...mean) (line 84-86).

Response: We appreciate the reviewer’s feedback on this point. We have re-framed the review more specifically around TDE and PCS (as opposed to TID). And we have tried to be more specific around what the review will provide, whilst also accepting the purpose scoping reviews have in providing an overview of the current literature. The revised text is now located in line 67-75.

Comment 10: Interesting I would have though you have enough of a rationale for a full PRISMA SR and a quality appraisal etc...but will read on (line 87-89)

Response: We appreciate the reviewer’s feedback on this point. We do however maintain our position and rationale for a scoping review, which we have tried to be clearer on throughout the paper.

Comment 11: In what? (line 91)

Response: We appreciate the reviewer’s highlighting this. We have made clear it is PCS research in soccer specific environments. This amendment can be seen in line 79.

Comment 12: A couple of things here. Just because this is the first of something doesn’t mean it's needed. This is language games we adopt because university admin who write REF impact cases ask us to use this language.

Second - I thought this review was around PCS and TID/TDE it now suggests soccer-specific environments (so does now include first team etc..?)

Third - The introduction is problematic for me at the moment. The absence of a clearly delineated and defined RQ or RQ's makes it difficult to establish what the MO for the review is. (line 93)

Response: We appreciate the reviewer’s feedback here. Regarding point one, we maintain this review is one of the first to explore PCS, soccer and links to TDE. However, hopefully the re-framing of the introduction now makes it clearer as to its need. Regarding point two, the review sought literature in soccer that had findings that could be applied to TDE environments/applied practitioners. For example, if there is research around how frequent players scan, then that paper may provide applications for practitioners working with players (exploratory nature of ScR). However, we have now made it more specific around TDE than TID. Thirdly, we agree the research question should have been introduced earlier, and so have brought it forward to line 81.

Comment 13: I wonder whether you could have achieved this with a full SR. A narrative analysis a quality appraisal would have given you a nice discussion. Not a deal-breaker for me but it just seems a little strange (line 98-100).

Response: We appreciate the comment raised. We maintain a scoping review was appropriate due to diverse methodologies even post inclusion/exclusion criteria rating and the aim to provide an overview of the PCS, soccer, TDE links and where researchers should look to further support applied practices. We have aimed to be more consistent with this rationale throughout and can be seen in line 89-95.

Comment 14: Which isn't stated (line 102)

Response: We appreciate the reviewer’s feedback on this point. As suggested, we brought the research question forward (line 81).

Comment 15: Ah...the RQ. This needs to go earlier after the introduction. This is different to the aim/objective presented earlier. This is a deal-breaker. What do you mean by TID/TDE outcome focused. This is vague and superficial and needs to be more precise. What type of soccer-specific environments (academy, schools, grassroot clubs, regional camps, university, college...I could go on). This needs to be delineated or defined. From reading the introduction you refer specifically to academy setting (Anglophile) - this seems much broader. Again I could be wrong. Your job is the give me precise information (line 105-107).

Response: We appreciate the reviewer’s concern on this point. As suggested, we have aimed to make the research question and focus of the review clearer, moved more specifically to TDE (than TID) and have aimed to make it clearer what we mean by this and the purpose of the review. These changes and refined clarity have been introduced in line 68-75.

Comment 16: Why 20 years. What's the rationale? Was this when a seminal PCS paper was published? (line 107).

Response: We appreciate the reviewer’s comment here. This was a review of the last 20 years due to the increased research interest in PCS since 2000 (Figure 1). However due to the time this paper was in holding, we have re-framed to examine 3 decades of evolving enquiry into PCS and soccer (line 78-81).

Comment 16: Not your fault but this was approaching four months ago. Can you re-run and establish whether any new articles could be added (line 109).

Response: We appreciate the reviewer’s comment here. We have re-run searches here to update to January 2025 and modified the report accordingly (line 101).

Comment 17: Any journal citation chaining used as a back stop? (line 111-112).

Response: We appreciate the reviewer’s feedback on this point. Although this would have been a sensible approach, we believe the coverage from four databases with additional handpicking process from google scholar help provide adequate coverage of relevant articles. Upon reflection, this is something that could have been included but has not been evidenced by other scoping reviews on similar areas (e.g., McCalman et al., 2021).

Comment 18: This alludes to the earlier point about the Introduction. If this is about PCS in a TID/TDE context then I would expect a more comprehensive argument to be presented (line 115-116).

Response: We appreciate the reviewer’s feedback on this point. We have included a clearer rationale for the inclusion of the ‘coach*’ search term in line 109-110.

Comment 19: If this review is about PCS and TID how come scouts was not included as a search term? (line 116).

Response: We appreciate the reviewer’s concern here. Following a re-framing around TDE, the rationale for incorporating scouts in this in now no longer required.

Comment 20: Wow. Enough for a SR.

Response: We appreciate the reviewer’s feedback here. After reviewing recent scoping reviews of a similar theme, we feel the number of search terms were in-line with other scoping review (e.g McCalman et al., 2021).

Comment 21: These are really vague. So I assume you are including literally all environments. There is no mention of gender (so assume male and female environments). I am still not sure I understand what applications/implications actually means? So if there is a coach perception of PCS in a talent setting this will be included. Does stakeholders extend to parents/teachers? These are not included in the search parameters and important if the environment is a school, college or university for example. How or where does the TID/TDE application appear in the study. Does the MO for the inclusion criteria mean a study must outline in the aim/objective TID/TDE? This is not clear. (line 132-136).

Response: We appreciate the reviewer’s feedback on these points. For point 1, we have aimed to be clearer on the inclusion criteria and rationale for this. Please see line 125-132. For point 2, we hope we have now made the approach to research applications clearer and can be seen in line 67-70. For point 3, line 127-130 highlight how the TDE link was searched for during screening.

Comment 22: In your final included studies you have a population that includes GK's does this not violate your inclusion criteria? (line 136)

Response: We appreciate the reviewer’s comment. We believe the framing of our criteria is suitable as the use of the goalkeeper in the study referred to is not the focus of the paper. The studies had to focus on outfield players and not include penalty kick analysis (line 131).

Comment 23: How was this resolved if JC and AM did not agree? (Line 142-144).

Response: We appreciate the reviewer’s feedback on this point. This sentence has been amended, reflected in line 139-140.

Comment 24: What data? Quants/Quals. Did you have a codebook to extract data and adopt another strategy? What are recordings? Can you again be precise with language (line 144-145).

Response: We appreciate the reviewer’s comment. We have re-framed some of the language used alongside introducing an analysis section to be clearer on how the papers were analysed (line 140-166).

Comment 25: What quality checks? Assume you don't mean quality appraisal here? (line 147)

Response: We appreciate the reviewer’s feedback on this point. We did not mean quality appraisal, and were referring to members of the research team reviewing articles selected/deselected to ensure the criteria set was being used appropriately.

Comment 26: Which is not clear (for me at least) (line 149).

Response: We appreciate the reviewer’s comment on this point. The data analysis section has been re-written and can be seen on line 148-166.

Comment 27: So are you only extracting quantitative data to establish associations between different variables? (line 150)

Response: We appreciate the reviewer’s comment on this point. This section has been re-written and can be seen in line 154-166.

Comment 28: Sorry to sound like a stuck record but again this is really vague. It doesn't really tell me anything. (line 150-152)

Response: We appreciate the reviewer’s comment on this point. This section has been re-written and can be seen in line 154-166.

Comment 29: And this for me is problematic. Even though it is a PRISMA ScR - it needs to be clearly defined (see Boland, Cherry and Dickson). I accept it can be broader in focus but it still requires a clearly defined question. If you maintain this position I would suggest you change the MO to a 'narrative review' rather than a PRISMA-ScR. (line 156).

Response: We thank the reviewer’s for their feedback. Line 146-150 continue to develop the rationale for the scoping review, alongside a more clearly defined research question than previous.

Comment 30: Was this decided post-hoc? (line 157)

Response: We appreciate the reviewer’s feedback here. Analysis of methodological approach quality was not undertaken with it not being a key feature of ScR.

Comment 31: This is not mentioned. This is another deal-breaker for me. What was the analysis? This needs to be outlined? (line 166)

Response: We appreciate the reviewer’s concern here. We have addressed this with the inclusion of an analysis section within the methodology (line 154-166).

Comment 32: Where did this theme originate form? Did you have a codebook where you deductively extracted a prior items/data? (line 171)

Response: We appreciate the reviewer’s feedback here. Hopefully the inclusion of the analysis section and more specific methods highlights how themes were generated (line 154-166).

Comment 33: Considering the MO for this review is PCS in TID/TDE there is not discussion around this here. I am not sure anyone working in TID/TDE would gain anything from reading this. (line 173-193)

Response: We appreciate the reviewer’s concern here. We have re-framed significant portions of the manuscript to reflect more specific links with TDE throughout.

Comment 34: It would be useful to *those included studies which you are referring to. (line 156)

Response: We appreciate the reviewer’s feedback on this point. As suggested, reviewed studies have been asterisked throughout.

Comment 35: You evidence from Australian youth academies above ob l.197 (line 201)

Response: We acknowledge the reviewer’s point here. We have re-written this paragraph to acknowledge the need for visual search research in younger age groups within TDE environments and can be reflected in line 212-229.

Comment 36: This has lurched to UK/Anglophile terminology. An audience outside of the UK/England will not know what youth development phase means. You have gone very specific to academy settings (assume in England). You seem to be introducing ideas in here that the review is not supporitng? Why not extend the point to other contexts? (line 201-205)

Response: We appreciate the reviewer’s feedback on this point. As suggested, we have moved the language award from UK terminology to support practitioners in a broader sense (e.g., line 221-224).

Comment 37: Wow. This is quite a leap. I am not sure you have the evidence to make this claim yet. It certainly will not help guide practitioners. Be specific. How will this guide practitioners?

---

## [Decision Letter · Decision Letter 1]

Perceptual-cognitive skills and talent development environments in soccer: A scoping review

PONE-D-24-10341R1

Dear Dr. Triggs,

We’re pleased to inform you that your manuscript has been judged scientifically suitable for publication and will be formally accepted for publication once it meets all outstanding technical requirements.

Kind regards,

Leonardo Vidal Andreato, PhD

Academic Editor

PLOS ONE

Additional Editor Comments (optional):

Reviewers' comments:

Reviewer's Responses to Questions

**Comments to the Author**

Reviewer #1: All comments have been addressed

2. Is the manuscript technically sound, and do the data support the conclusions?

Reviewer #1: Yes

3. Has the statistical analysis been performed appropriately and rigorously?

Reviewer #1: N/A

4. Have the authors made all data underlying the findings in their manuscript fully available?

Reviewer #1: Yes

5. Is the manuscript presented in an intelligible fashion and written in standard English?

Reviewer #1: Yes

Reviewer #1: Dear author,

Thank you for your the revisions and the amendments to the scoping review. The Introduction is tighter and the presentation of a broader research question is in-line with a scoping review. In terms of future work (i.e. if you intend to develop the PCS angle to a full SR) then I would consider registering a protocol with either OSF or publishing the protocol initially. This will allow you to deal with some of the thorny methodological and rigour related issues (publication bias, grey literature, blind reviewing, post-hoc analysis etc...) included in this review. My recommendation to the EiC will be that you have responded to all the suggestions and where you have 'pushed back' there are robust and reasoned justifications.

**Do you want your identity to be public for this peer review?** For information about this choice, including consent withdrawal, please see our Privacy Policy

Reviewer #1: No

---

## [Editor Report · Acceptance letter]

PONE-D-24-10341R1

PLOS ONE

Dear Dr. Triggs,

I'm pleased to inform you that your manuscript has been deemed suitable for publication in PLOS ONE. Congratulations! Your manuscript is now being handed over to our production team.

Kind regards,

on behalf of

Dr. Leonardo Vidal Andreato

Academic Editor

PLOS ONE